# The Metabolic Activation of Sofosbuvir Is Impaired in an Experimental Model of NAFLD

**DOI:** 10.3390/biology11050693

**Published:** 2022-04-30

**Authors:** Daniela Gabbia, Marco Roverso, Samantha Sarcognato, Ilaria Zanotto, Nicola Ferri, Francesco Paolo Russo, Maria Guido, Sara Bogialli, Sara De Martin

**Affiliations:** 1Department of Pharmaceutical and Pharmacological Sciences, University of Padova, 35131 Padova, Italy; daniela.gabbia@unipd.it (D.G.); ilaria.zanotto.1@studenti.unipd.it (I.Z.); 2Department of Chemical Sciences, University of Padova, 35131 Padova, Italy; marco.roverso@unipd.it (M.R.); sara.bogialli@unipd.it (S.B.); 3Department of Medicine, University of Padova, 35131 Padova, Italy; samantha.sarcognato@gmail.com (S.S.); nicola.ferri@unipd.it (N.F.); mguido@unipd.it (M.G.); 4Department of Surgery, Oncology and Gastroenterology, University of Padova, 35131 Padova, Italy; francescopaolo.russo@unipd.it

**Keywords:** NAFLD, sofosbuvir, HCV, drug metabolism, pharmacokinetics, UMP-CMPK1

## Abstract

**Simple Summary:**

Steatosis is a disease of the liver characterized by the deposition of lipids in hepatocytes. Although the effect of steatosis on the metabolism of drugs has been investigated, the findings obtained so far are still controversial. We here evaluated the pharmacokinetics of the main inactive metabolite of sofosbuvir, the first direct antiviral agent reaching the market for hepatitis C treatment, called GS-331007. We demonstrated that plasmatic levels of GS-331007 increased significantly in rats with steatosis, whereas the expression of the enzyme UMP-CMPK, responsible for the activation of sofosbuvir to its active metabolite GS-331007-TP, was significantly lower than that of healthy animals. The reduction of UMP-CMPK expression suggests an impairment of sofosbuvir activation, giving a possible explanation for the reduction of its efficacy in patients affected by genotype 3 HCV, which is often associated with liver steatosis.

**Abstract:**

The effect of liver steatosis on drug metabolism has been investigated in both preclinical and clinical settings, but the findings of these studies are still controversial. We here evaluated the pharmacokinetic profile of the main sofosbuvir metabolite GS-331007 in healthy animals and rats with non-alcoholic fatty liver disease (NAFLD) after the oral administration of a single 400 mg/kg dose of sofosbuvir. The plasma concentration of GS-331007 was evaluated by HPLC-MS. The expression of the two enzymes uridine monophosphate-cytidine monophosphate kinase 1 (UMP-CMPK1), and nucleoside diphosphate kinase (ND-PK), responsible for the formation of the active metabolite GS-331007-TP, were measured by qRT-PCR and Western Blot. We demonstrated that in rats with steatosis, the area under the plasma concentration-vs-time curve (AUC) and the peak plasma concentration (C_max_) of GS-331007 increased significantly whereas the expression of UMP-CMPK was significantly lower than that of healthy animals. The reduction of UMP-CMPK expression suggests an impairment of sofosbuvir activation to GS-331007-TP, giving a possible explanation for the reduction of sofosbuvir efficacy in patients affected by genotype 3 Hepatitis C virus (HCV), which is often associated with liver steatosis. Furthermore, since GS-331007 plasma concentration is altered by steatosis, it can be suggested that the plasma concentration of this metabolite may not be a reliable indicator for exposure-response analysis in patients with NAFLD.

## 1. Introduction

Hepatitis C virus (HCV) is one of the major causes of chronic liver disease, culminating in cirrhosis, hepatocellular carcinoma (HCC) and increased mortality from both hepatic and extrahepatic disorders. It is estimated that nearly 400,000 people die every year because of the consequences of this viral infection [1]. Phylogenetic analyses of viral genomic sequences have identified at least 6 major HCV genotypes [2], among which HCV 1, 2 and 3 are widely disseminated and have been thoroughly assessed for their epidemiology, natural disease history and treatment outcomes. A strong association between liver steatosis and genotype 3 HCV infection has been noticed since the diagnosis of steatosis in these patients is 5-fold more frequent compared with non-3 HCV genotypes [3]. Accordingly, it is considered that in this setting steatosis is of viral origin and is called “viral steatosis” [4]. The existence of a complex interplay between metabolic diseases, fatty liver and chronic viral hepatitis has been reported both for HCV and hepatitis B virus (HBV) [5], leading to the observation that a multidisciplinary approach should be used for patients affected by these conditions, with careful and personalized decisions for their pharmacological treatment. Furthermore, the effect of liver disease, including non-alcoholic fatty liver disease (NAFLD) and its complication non-alcoholic steatohepatitis (NASH), on the metabolism of drugs, especially cytochrome P450 (CYP)-mediated phase 1 reactions [6], has been investigated in both preclinical [7] and clinical studies [8,9]. Furthermore, it has been demonstrated that nutrition, in particular a diet rich in fat and fructose (HFHF diet) [10], which represents one of the most used strategies to obtain experimental models of liver steatosis, might have a significant impact on the expression of genes involved in drug metabolism [11,12].

The discovery of direct antiviral agents (DAAs) had a dramatic impact on the therapeutic management of HCV [13]. Sofosbuvir, a second-generation nucleotide polymerase inhibitor, is a nucleotide prodrug that undergoes intracellular metabolism to form the pharmacologically active uridine triphosphate analog GS-331007-TP (Figure 1). After oral administration, sofosbuvir reaches the liver by the portal vein and penetrate hepatocytes, where it is immediately hydrolyzed by the two enzymes carboxylesterase 1 and cathepsin A to form the Metabolite X. These metabolic steps are followed by further metabolism to form GS-331007 mono-(GS-331007-MP), di- and triphosphate (GS-331007-TP), the last being the active compound [14], by the enzymes histidine-triad nucleotide binding protein 1 (HINT1), uridine monophosphate-cytidine monophosphate kinase 1 (UMP-CMPK1), and nucleoside diphosphate kinase (NDPK), respectively. The three phosphorylated metabolites are negatively charged and cannot pass the cell membrane of hepatocytes. The monophosphate can be dephosphorylated to the nucleoside GS-331007, which is devoid of antiviral activity against HCV [15]. Since none of the cellular pyrimidine nucleoside kinases can re-phosphorylate GS-331007, this metabolite is released into systemic circulation and undergoes renal excretion [14]. GS-331007 has been considered the primary analyte of interest in clinical pharmacokinetic (PK) studies of sofosbuvir and was used for exposure-response analysis in conjunction with its parent drug in Phase III trials.

In the light of these considerations, since liver steatosis is strongly associated with HCV genotype 3 infection, which is more resistant to sofosbuvir antiviral action than other genotypes [16] and alteration of hepatic drug metabolism have been reported in animals and patients with NAFLD [7,8], we hypothesized that steatosis may affect sofosbuvir enzymatic activation in the liver and consequently its efficacy. Therefore, the aim of this study was to ascertain whether NAFLD, obtained by the administration of a HFHF diet [10,17], was associated with changes in the PK parameters of the sofosbuvir main metabolite, GS-331007.

## 2. Materials and Methods

### 2.1. Animal Study and Histological Analysis

All the experimental procedures involving animals were conducted in compliance with national and international guidelines for the handling and use of animals used for scientific purposes (Authorization no. 721-2017-PR, 2 October 2017) and appropriately designed to minimize their pain or discomfort. All the animals were maintained under controlled conditions, with a 12/12 h dark/light cycle and free access to food and drink. To obtain the animal model of NAFLD, 6 male Sprague Dawley rats (body weight 150–170 g, age 6 ± 1 weeks) were fed for 12 weeks with a diet enriched in fat (high fat diet, HFD, 60% Fat, kcal from: 23.5% protein, 18.4% carbohydrate; 60.3% fat; Altromin; Germany) boosted with 30% of fructose in drinking water. Six rats fed with standard diet were used as controls. After 12 weeks, a single 400 mg/kg sofosbuvir dose was administered to the rats by oral gavage, after an overnight fast. To perform the PK analysis, plasma samples have been collected from the caudal vein before and 0.5, 1, 2, 3, 4, 6 and 24 h after the administration of sofosbuvir. The rats were sacrificed after 24 h by an excess of isoflurane anesthesia (4.5% in oxygen), and their livers were collected for histological and biochemical analyses. Blood was collected by cardiac puncture to measure the plasma concentration of biochemical liver function markers, such as alanine aminotransferase (ALT), aspartate aminotransferase (AST), and triglycerides. These analyses were performed by the Central Laboratory of the University Hospital of Padova, using standard techniques [18]. The histological evaluation of liver steatosis was performed on hematoxylin and eosin (H&E)-stained slices, by a pathologist who was blinded to the diet administered to the rats [11].

### 2.2. Quantification of GS-331007 Plasma Concentration by HPLC-MS

In total, 400 μL of cold acetonitrile spiked with 50 µg/L of internal standard (Trimethylamine-d9 N-Oxide (TMAO-d9), Sigma Aldrich, Italy) were added to 100 μL of plasma, and samples were then centrifuged for 10 min at 14,000 rpm and 25 °C (Hettich Mikro 120 Benchtop Centrifuge). Two microliters of the supernatant were injected into the LC-HRMS detection system equipped with an Ultimate 3000 UHPLC chromatograph coupled with a QExactive hybrid quadrupole-Orbitrap mass spectrometer (Thermo Fisher Scientific; Waltham, MA, USA). The analytical column was a Kinetex 2.6 μm EVO C18, 100 A, 100 × 2.1 mm (Phenomenex, Bologna, Italy), thermostated at 25 °C. The mobile phase components A and B were water and acetonitrile, respectively, both containing 1 mM ammonium fluoride. The eluent flow rate was 0.25 mL/min. The mobile phase gradient profile was (t in min) t_0−3_ 0% B; t_3−18_ 0–100% B, t_18−23_ 100% B; t_23−24_ 0% B and the six minutes for the re-equilibrium. The MS conditions were set as follows: electrospray (ESI) ionization in positive mode, resolution 35,000, AGC target 3 × 10^6^, max injection time 50 ms and scan range 50–750 a.m.u. The capillary voltage was 3.5 kV, capillary temperature was 320 °C, auxiliary gas was nitrogen at 40 a.u. Mass calibration was performed with the standard Pierce ESI Positive Ion Calibration Solution (Thermo Fisher Scientific). The MS data were analyzed with the Xcalibur 4.0 software (Thermo Fisher Scientific). Quantification was carried out by external calibration by using a five-point calibration curve obtained by spiking blank plasma with sofosbuvir and GS331007 in the range 530–5.3 µg/L and 260 µg/L–2.6 ug/L (1–0.01 µM), respectively. Linearity showed an *R*^2^ > 0.99 for both analytes. Limits of detection (LOD) of the method were 0.3 µg/L for sofosbuvir and 1.3 µg/L for GS331007).

### 2.3. Quantification of Gene Expression by Means of qRT-PCR

Total RNA was extracted from liver tissue by means of the commercial Total RNA SV isolation kit (Promega Corporation; Madison, WI, USA). The gene expression of UMP-CMPK and NDPK was measured using a one-step commercial kit (Takara Bio; San Jose, CA, USA) and the Eco Real Time PCR system (Illumina; San Diego, CA, USA), as already reported [11]. The relative mRNA expression was calculated according to the ∆∆Ct method [19], using *β*-actin as housekeeping gene. The sequences of the primers used in this study are reported in Table 1.

### 2.4. Quantification of Protein Expression by Western Blot

The protein expression of UMP-CMPK1 was assessed by Western blot, according to a previously described method [20]. Briefly, hepatic tissue lysates were prepared by homogenizing frozen liver tissue in cold RIPA buffer with Ultraturrax for 10 s and their protein concentration was quantified by means of the Pierce BCA Protein Assay kit (Thermo Fisher Scientific), following the manufacturer’s instructions. For the electrophoretic run, 30 μg of proteins per lane were subjected to SDS-PAGE and then blotted onto a nitrocellulose membrane by means of a Biorad Turbo-blot system (Biorad Laboratories S.r.l., Milan, Italy). After a blocking step performed with 10% skim milk in TBS-T, the membrane was incubated overnight at 4 degrees with a primary anti-UMP-CMPK1 antibody (Santa Cruz Biotechnology; Dallas, TX, USA, diluted 1:500), and then with a secondary anti-mouse HRP-conjugated antibody (LGC Seracare; Milford, MA, USA, diluted 1:100). A rabbit anti-GAPDH antibody (Santa Cruz, dilution 1:1000) was used to quantify the protein expression of GAPDH, used as loading control. Reactive proteins were stained with the Luminata Classico Western HRP Substrate (Merck-Millipore; Milan, Italy) and visualized with an Azure c400 Gel Imaging System (Azure Biosystems; Dublin, CA, USA).

### 2.5. Pharmacokinetic Analysis

The PK parameters have been calculated by means of the Microsoft Excel plug-in PK Solver, using standard formulae [21]. In detail, since the data we obtained could not be fitted to any poly-exponential equation, model-independent PK parameters were evaluated. The peak plasma concentration (C_max_) and the time to C_max_ (T_max_) were the observed values. The area under the plasma drug concentration–vs-time curve (AUC) was calculated up to 24 h and then extrapolated to infinity.

### 2.6. Statistical Analysis

The data have been analyzed using the software GraphPad Prism 8.0 (GraphPad Software Inc.; San Diego, CA, USA) ver. 8.0. The reported results are obtained from three independent experiments and, unless otherwise stated, are expressed as mean ± standard error of the mean (S.E.M.). Comparison of the results obtained from controls and rats with NAFLD has been performed by means of Student’s t test for unpaired data. *p* < 0.05 was considered statistically significant.

## 3. Results

### 3.1. The Administration of HFHF Diet Causes Liver Steatosis in Rats and Affects Liver Function

Rats fed for 12 weeks with a diet rich in fat boosted with 30% in tap drinking water developed liver steatosis, as shown in Figure 2. In fact, histological examination of the liver in this group of rats showed a diffuse macro- and medio-vesicular steatosis, involving the whole liver lobule from peri-venular to peri-portal zones (Figure 2B). None of the rats in the control group fed with a normal diet developed steatosis and showed a normal liver histology (Figure 2A). Accordingly, a significant increase in plasma triglycerides was observed in animals with NAFLD. An impairment of liver function in these animals is indicated by the significant increase in the plasma concentration of ALT (Figure 2C).

### 3.2. Liver Steatosis Affects the Pharmacokinetic Parameters of GS331007

Table 2 and Figure 3 show that NAFLD has a deep impact on the pharmacokinetics of the main sofosbuvir metabolite GS-331007, since a significant increase in its area under the plasma concentration-vs-time curve (AUC) and peak plasma concentration (C_max_) was observed in rats with hepatic steatosis, as well as a decrease in time to C_max_ (T_max_).

Considered the well-known metabolic routes of sofosbuvir [14,15], to better understand the impact of NAFLD on the metabolism and activation of this drug, we measured the mRNA expression of the two enzymes involved in the synthesis of the active triphosphate drug GS-330117-TP, i.e., UMP-CMPK1 and ND-PK, and the protein expression of UMP-CMPK1. Interestingly, the mRNA (Figure 4A) and protein expression (Figure 4C,D) of UMP-CMPK was significantly reduced (*p* < 0.05) in rats with NAFLD, which can at least in part explain the increase in GS-331007 plasma concentration, obtained by the dephosphorylation of the monophosphate intermediate, since its UMP-CMPK-dependent metabolic route to the active triphosphate metabolite is reduced. NAFLD had no impact on the mRNA expression of ND-PK (Figure 4B).

## 4. Discussion

The impact of liver dysfunction on the metabolism of drugs has been long studied, manly focusing on cirrhosis of different origins [6]. However, also metabolic disorders such as NAFLD, besides affecting liver physiology, may influence several mechanisms which modulate drug metabolism, such as inflammation and oxidative stress [22,23,24], indicating the need for the optimization of the clinical dose in patients with such disorders [25]. The main part of the current knowledge about the effect of NAFLD on drug metabolism is based on preclinical observations which clinical translation is poor. In addition, even focusing only on preclinical animal studies, the effect of NAFLD on drug metabolism is still controversial and lacks a consensus among different reports. However, recent findings suggest that NAFLD may cause alterations in drug metabolizing enzymes although, as stated before, only a limited part of these results are consistent across studies and species. Apart from the downregulation of CYP3A [11] and upregulation of CYP2E1 [26], which have been consistently demonstrated, the results regarding other enzymes are either lacking or conflicting [27]. There is a consensus that oxidative stress and inflammatory mediators such as TNF-α and IL-6, which are hallmarks of liver steatosis, can alter the function of nuclear receptors (NRs) in NAFLD [28,29]. Interestingly, NRs such as Constitutive Androstane Receptor (CAR) and Pregnane X Receptor (PXR) are transcription factors modulating the expression of drug metabolizing enzymes, including different CYP isoforms [6]. However, to our knowledge, a systematic analysis of the effect of NAFLD on the enzymes responsible for sofosbuvir activation and metabolism has never been performed. Of note, sofosbuvir cannot be detected in rat plasma, in accordance with what observed by Wang and collaborators [30], who demonstrated that sofosbuvir is rapidly degraded in rodent plasma, due to the high levels of esterase activity. In this work, we demonstrated that the plasma concentration of the sofosbuvir metabolite GS-331007 is significantly increased in rats with NAFLD, in the liver of which the expression of the enzyme UMP-CMPK shows a significant drop compared to healthy animals. Since UMP-CMPK catalyzes the first reaction leading to the formation of the active metabolite GS-331007-TP, we hypothesize a role of this enzyme in the observed pharmacokinetic profile of GS-331007, i.e., the observed increase in GS-331007 is due to the reduction of UMP-CMPK-mediated GS-331007-MP phosphorylation. In this case, the dephosphorylation of GS-331007-MP to GS-331007 may become a preferential metabolic route, leading to the formation of an uncharged intermediate, which can be easily eliminated by the hepatocyte. It should be noticed that GS-331007 and sofosbuvir have been used for exposure-response analysis in sofosbuvir Phase III trials. This is because the drug-related material-based calculations conducted during the clinical development by analyzing the AUC of sofosbuvir and each metabolite that was detectable in plasma, demonstrated that the predominant circulating species after sofosbuvir administration was GS-331007 (90% of drug-related material) [14]. Our data indicate that sofosbuvir metabolic activation and GS-331007 plasma concentration are altered by steatosis in rats, thereby suggesting that GS-331007 plasma concentration may not be a reliable indicator for exposure-response analysis in patients with NAFLD. The translational value of our results needs to be confirmed by clinical studies, investigating sofosbuvir pharmacokinetics in patients selected according to the diagnosis of NAFLD. Furthermore, further studies are encouraged to analyze in detail the effect of liver steatosis on the metabolism and disposition of drugs and their efficacy, considering the increasing burden of NAFLD and NASH [31]. It should be noticed that our study does not investigate the effect on the hepatic enzymes involved in sofosbuvir metabolism exerted by a prolonged administration of this drug, which is the therapeutic regimen used in clinical practice. Furthermore, sofosbuvir is usually not administered as a monotherapy, but in combination with other DAAs, such as velpatasvir and voxilaprevir [32], which are metabolized by CYP enzymes (CYP2B6, CYP2C8 and CYP3A4) and are known inhibitors of drug transporters, such as p-glycoprotein, breast cancer resistance protein (BCRP), and some solute-carrier transporters [32]. Therefore, the sofosbuvir exposure of HCV patients in the real-world practice is surely dependent on mechanisms that are more complex than in this study. However, we demonstrated for the first time that NAFLD can modulate the expression of an enzyme which is pivotal for sofosbuvir activation.

## 5. Conclusions

In this work, we demonstrated that the main sofosbuvir metabolite GS-331007 is significantly increased in rats with NAFLD after sofosbuvir administration, due to the altered expression of an enzyme responsible for sofosbuvir metabolism. Furthermore, the reduction of UMP-CMPK expression suggests an impairment of the activation of sofosbuvir to its active metabolite GS-331007-TP, giving a possible explanation for the reduction of sofosbuvir efficacy in patients affected by genotype 3 HCV, although this requires clinical confirmation. In general, this finding suggests that NAFLD can be responsible for pharmacokinetic changes that should be monitored to avoid therapeutic failure or unexpected toxicity.

## Figures and Tables

**Figure 1 biology-11-00693-f001:**
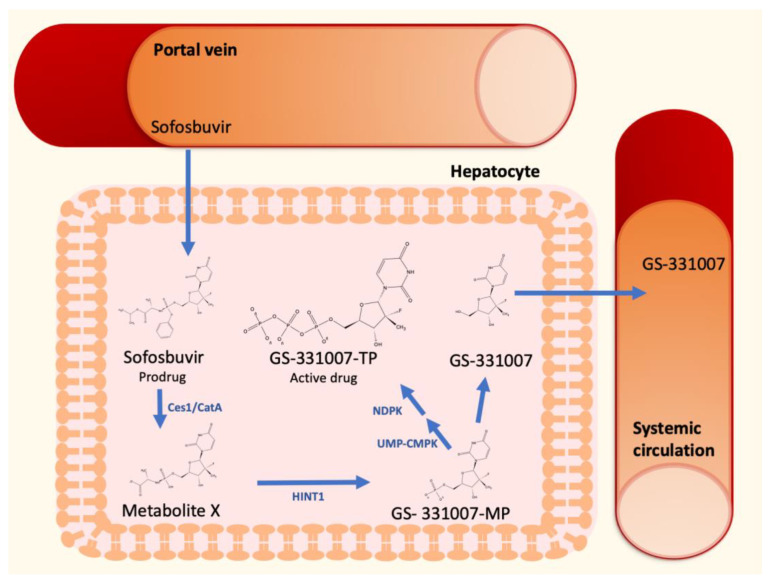
Sofosbuvir hepatic metabolism and activation to GS-331007-TP.

**Figure 2 biology-11-00693-f002:**
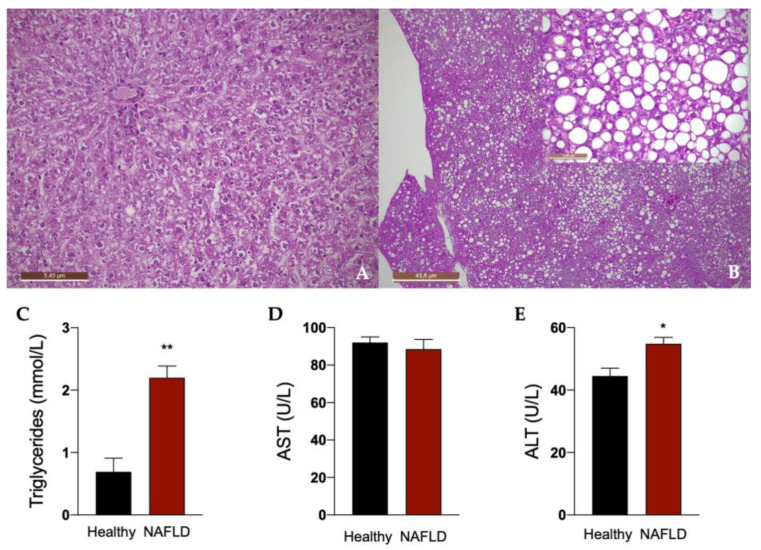
Liver histology of rats. Representative photomicrographs of liver tissue from a rat fed with standard (**A**) and HFHF (**B**) diet (hematoxylin and eosin–H&E staining). 2.5× (**A**,**B**), 20× (insert of B) Magnification. The histological was performed in 6 healthy animals and 6 rats with NAFLD. Plasma concentrations of triglycerides (**C**), aspartate transaminase (AST, **D**) and alanine transaminase (ALT, **E**). Results are reported as means and S.E.M. of 6 animals per group. * *p* < 0.05, ** *p* < 0.01 vs. healthy rats.

**Figure 3 biology-11-00693-f003:**
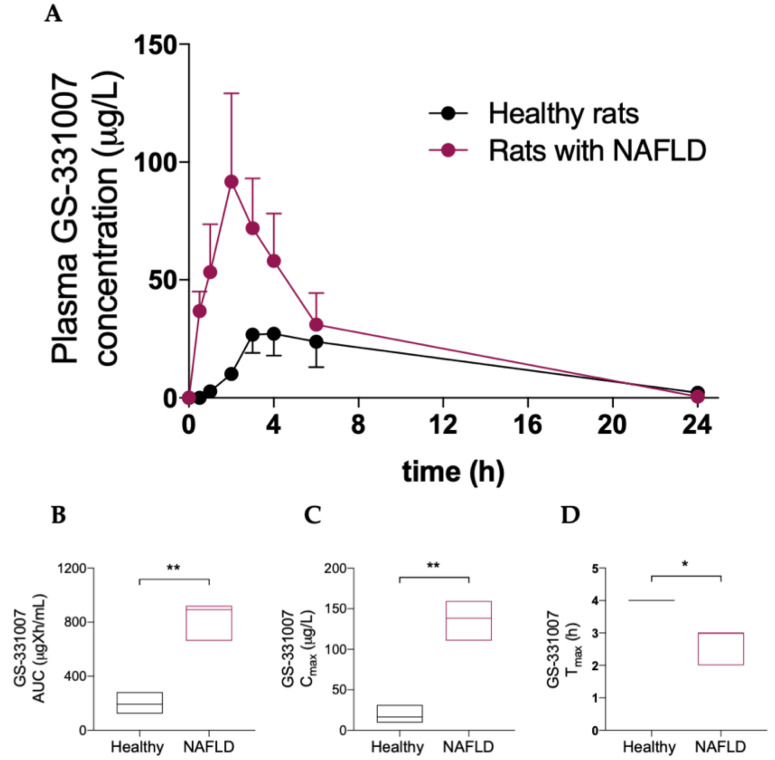
(**A**) Plasma concentration vs. time curve of GS331007. Box plots of GS331007 AUC (**B**), C_max_ (**C**) and T_max_ (**D**)**.** The horizontal line plots the median. Results are reported of means and S.E.M. of 6 animals per group. * *p* < 0.05, ** *p* < 0.01 vs. healthy rats.

**Figure 4 biology-11-00693-f004:**
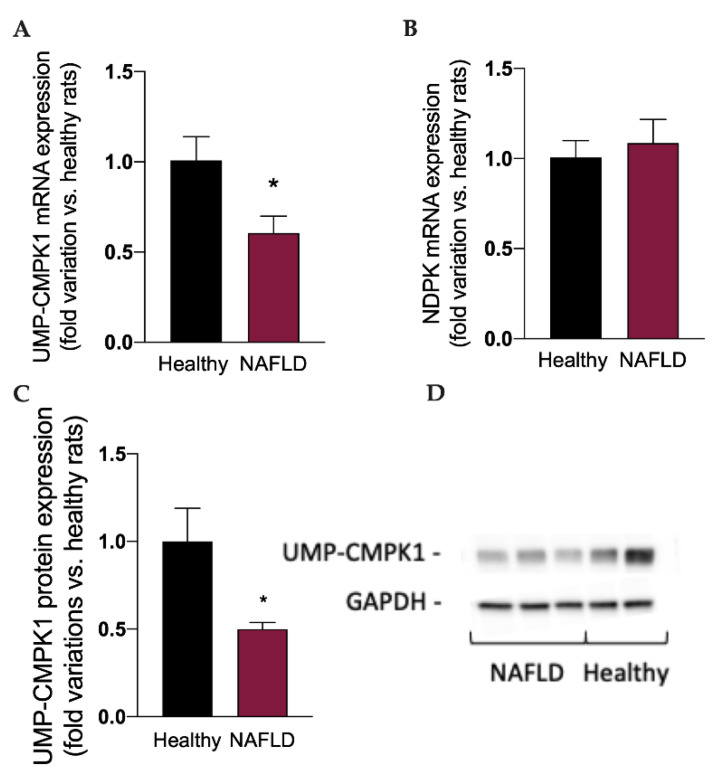
mRNA expression of the enzymes UMP-CMPK1 (**A**) and NDPK (**B**), responsible for the activation of sofosbuvir to the active compound GS-331007-TP. Protein expression of the enzyme UMP-CMPK1 (**C**). A representative Western Blot is shown in (**D**). Results are reported as means and S.E.M. of 6 animals per group. * *p* < 0.05 vs. healthy rats.

**Table 1 biology-11-00693-t001:** Sequences of the Primers Used in the Study for qRT-PCR Experiments.

Target	Forward	Reverse	Product Size (bp)
UMP-CMPK1	ATGAAGCCGTTGGTCGTGT	GCAGAAAGGTGTGTGTAGCCA	101
NDPK	CGACTACACTTCTTGCTTCTGC	GGAACCCCTTCTGCTCGAAT	128
*β*−actin	GCCACCAGTTCGCCATGGA	TTCTGACCCATACCCACCAT	163

**Table 2 biology-11-00693-t002:** Pharmacokinetic Parameters of GS-331007 in Healthy Rats and Rats with NAFLD. Data are Reported as Mean ± SD (AUC, C_max_) or Median (Range) (T_max_).

PK Parameter	Healthy Rats	Rats with NAFLD
AUC (μg × h/mL)	199.7 ± 79.52	825.4 ± 141.8
C_max_ (μg/L)	19.15 ± 11.18	136.2 ± 24.58
T_max_ (h)	4 (0)	3 (1)

## Data Availability

The data used in this study are at disposal by the corresponding author upon reasonable request.

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
