# Peer review of "The Metabolic Activation of Sofosbuvir Is Impaired in an Experimental Model of NAFLD"

_biology, 2022, doi:10.3390/biology11050693_

Round 1

Reviewer 1 Report

The manuscript entitled: “The metabolic activation of sofosbuvir is impaired in an experimental model of NAFLD ” aimed to evaluate the influence of non-alcoholic fatty liver disease on the metabolism of sofosbuvir. The manuscript is interesting and well written. Some aspects should be clarified before publication.

  1. Please specify the drugs used for anesthesia and the relevant doses per body weight.
  2. All the abbreviations should be defined at the first use in the text
  3. Please specify all the manufacturers for the antibodies used.
  4. In the results you specify the quantification of ALT, AST and Triglycerides. Please add in the material and methods the methods used for these determinations.
  5. In the histological figures can you indicate by arrows where the changes have a place.
  6. Figure 3 B, C, D should be changed as they are very small and can not be easily read
  7. In discussion add the limitations of the study

Author Response

The manuscript entitled: “The metabolic activation of sofosbuvir is impaired in an experimental model of NAFLD ” aimed to evaluate the influence of non-alcoholic fatty liver disease on the metabolism of sofosbuvir. The manuscript is interesting and well written. Some aspects should be clarified before publication.

1. Please specify the drugs used for anesthesia and the relevant doses per body weight.

We added the name of the anesthetic drug and its dose in the text.

2. All the abbreviations should be defined at the first use in the text

We thank the Reviewer for this observation and modified the text accordingly.

3. Please specify all the manufacturers for the antibodies used.

We added this information in the text.

4. In the results you specify the quantification of ALT, AST and Triglycerides. Please add in the material and methods the methods used for these determinations.

We added this information in the text.

5. In the histological figures can you indicate by arrows where the changes have a place.

We changed Fig. 2 in order to make clearer the development of steatosis.

6. Figure 3 B, C, D should be changed as they are very small and cannot be easily read

We thank the reviewer for this observation and modified the Figs. accordingly.

7. In discussion add the limitations of the study

We thank the reviewer for this suggestion and modified the discussion accordingly.

Reviewer 2 Report

The study conducted by Gabbia D. et al. is concerned with clarifying an aspect, so far little investigated, related to the metabolism of sofosbuvir and the possible interactions with the enzymatic activity in animal model of NAFLD.

As for Sofosbuvir, like many drugs, the in vivo metabolism of phosphate, phosphonate, and phosphinate prodrugs is impacted by their enzymatic activation. Because diffusion usually happens faster than payload release and reaches equilibrium, the enzymatic payload release functions as the rate limiting step of cell entry. The author motivates the interest in this specific topic of antiviral pharmacokinetics as it is known the slight reduction in response to DAA treatment of patients with genotype 3. In particular this genotype has been associated with an "induced viral steatosis". 
According to a correct and reliable methodology, the authors come to the conclusion that after a single dose of sofosbuvir in the mouse animal model with induced steatosis, the levels of the non-active metabolites of the drug are higher in NAFLD mice than in the "wild". Similar findings in terms of reduced antiviral efficacy in NAFLD patients have been reported in patients with HBV treated with tenofovir. In these clinical studies, a reduced processing capacity from prodrug to active metabolites was found in patients with NAFLD compared to treated subjects. 
The study of Gaggia D. et al. has the validity of demonstrating for the first time a reduced effect of transformation into active metabolites in patients treated with sofosbuvir and NAFLD, and it will enhance future clinical research. However, their results suffer from some limitations that at the moment do not allow to arrive at the conclusion that the reduced capacity of NAFLD liver to trasform sofosbuvir in its active metabolite, is the main explanation of its reduced efficacy.
In particular, their analysis is limited to a single dose of sofosbuvir. It cannot exclude that the inductive effect of sofosbuvir on the enzymatic activity, does not allow a further augmentation of the active metabolite availability, also in NAFLD rats.

Author Response

The study conducted by Gabbia D. et al. is concerned with clarifying an aspect, so far little investigated, related to the metabolism of sofosbuvir and the possible interactions with the enzymatic activity in animal model of NAFLD.

As for Sofosbuvir, like many drugs, the in vivo metabolism of phosphate, phosphonate, and phosphinate prodrugs is impacted by their enzymatic activation. Because diffusion usually happens faster than payload release and reaches equilibrium, the enzymatic payload release functions as the rate limiting step of cell entry. The author motivates the interest in this specific topic of antiviral pharmacokinetics as it is known the slight reduction in response to DAA treatment of patients with genotype 3. In particular this genotype has been associated with an "induced viral steatosis". 
According to a correct and reliable methodology, the authors come to the conclusion that after a single dose of sofosbuvir in the mouse animal model with induced steatosis, the levels of the non-active metabolites of the drug are higher in NAFLD mice than in the "wild". Similar findings in terms of reduced antiviral efficacy in NAFLD patients have been reported in patients with HBV treated with tenofovir. In these clinical studies, a reduced processing capacity from prodrug to active metabolites was found in patients with NAFLD compared to treated subjects. 
The study of Gaggia D. et al. has the validity of demonstrating for the first time a reduced effect of transformation into active metabolites in patients treated with sofosbuvir and NAFLD, and it will enhance future clinical research. However, their results suffer from some limitations that at the moment do not allow to arrive at the conclusion that the reduced capacity of NAFLD liver to transform sofosbuvir in its active metabolite, is the main explanation of its reduced efficacy.
In particular, their analysis is limited to a single dose of sofosbuvir. It cannot exclude that the inductive effect of sofosbuvir on the enzymatic activity, does not allow a further augmentation of the active metabolite availability, also in NAFLD rats.

We thank the reviewer for this clear and useful comment. We agree on the fact that sofosbuvir metabolism in a clinical setting, and even more in a real-world setting, is regulated by a panel of factors which are difficult to predict. Regarding the point he raised, to our knowledge, no information about a possible induction of UMP-CMPK by sofosbuvir treatment has been reported. We added a paragraph in the discussion, in order to discuss these points ad indicate them as study limitations.

Reviewer 3 Report

Gabbia et al demonstrated main sofosbuvir metabolite, GS-331007, greatly increased in rats with NAFLD after sofosbuvir administration through some typical techniques, suggesting NAFLD can be responsible for pharmacokinetic further. Although the work seems to be interesting, some gaps need to consider to address

Although the work seems to be interesting, some gaps need to consider to address further.
1. Introduction: the relationship between NAFLD and hepatitis that are not well introduced
2. Figure 1 needs to revise further (with a liver, maybe) to summarize their finding in the work, making its a graphical abstract for further attraction. 
3. Figure 4B,  protein expression needs to provide clearer immunoblotting, especially NAFLD and Healthy's lands were not much different; and supplement the blot before cropping with a ladder to make sure it worked. 
4. Each experiment needs to perform three-time repeatedly at least. So, "4.5. Pharmacokinetic and statistical analyses" needs to present separately. A new chapter to mention their statistical analyses must be provided clearly.
5. Figure 2A, B is poor to demonstrate their purpose. Liver of rats from A) control group showing the normal histological structure of hepatic lobules, B) the-treated group showing the cell activation... All must provide a larger magnification to show focal necrosis of hepatocytes, as well as apoptosis of hepatocytes, regards the treatment. Also, a scale bar is needed.

6. The manuscript still needs to check thoroughly English editing

Author Response

Gabbia et al demonstrated main sofosbuvir metabolite, GS-331007, greatly increased in rats with NAFLD after sofosbuvir administration through some typical techniques, suggesting NAFLD can be responsible for pharmacokinetic further. Although the work seems to be interesting, some gaps need to consider to address

Although the work seems to be interesting, some gaps need to consider to address further.
1. Introduction: the relationship between NAFLD and hepatitis that are not well introduced

We thank the reviewer for this observation and modified the text accordingly.

2. Figure 1 needs to revise further (with a liver, maybe) to summarize their finding in the work, making its a graphical abstract for further attraction. 

We have prepared a graphical abstract reporting a general picture of the meaning of the work.

3. Figure 4B,  protein expression needs to provide clearer immunoblotting, especially NAFLD and Healthy's lands were not much different; and supplement the blot before cropping with a ladder to make sure it worked. 

We improved the quality of Fig. 4 and added the image of the whole membrane to ascertain the goodness of the WB analysis.

4. Each experiment needs to perform three-time repeatedly at least. So, "4.5. Pharmacokinetic and statistical analyses" needs to present separately. A new chapter to mention their statistical analyses must be provided clearly.

We thank the Reviewer for this observation and modified the text accordingly.  

5. Figure 2A, B is poor to demonstrate their purpose. Liver of rats from A) control group showing the normal histological structure of hepatic lobules, B) the-treated group showing the cell activation... All must provide a larger magnification to show focal necrosis of hepatocytes, as well as apoptosis of hepatocytes, regards the treatment. Also, a scale bar is needed.

As described in the text, the animals in the treated group only developed steatosis. We did not observe hepatocyte necrosis or apoptosis. Pictures have been re-done at higher magnification and a scale bar has now been included

6. The manuscript still needs to check thoroughly English editing

Our manuscript was checked by an English mother tongue. We hope that is now suitable for publication.

Round 2

Reviewer 1 Report

The authors done all my comments and the manuscript is ready for acceptance. My decision is accept.

Reviewer 3 Report

All requests/reviews/suggestions are made. Thank you for revising the work thoroughly